# Colossal and reversible barocaloric effect in liquid-solid-transition materials *n*-alkanes

Jianchao Lin [1], Peng Tong [1✉], Kai Zhang[1], Kun Tao[1], Wenjian Lu [1✉], Xianlong Wang [1✉], Xuekai Zhang[1], Wenhai Song[1] & Yuping Sun [1,2]

Emerging caloric cooling technology provides a green alternative to conventional vapor-compression technology which brings about serious environmental problems. However, the reported caloric materials are much inferior to their traditional counterparts in cooling capability. Here we report the barocaloric (BC) effect associated with the liquid-solid-transition (L-S-T) in *n*-alkanes. A low-pressure of ~50 MPa reversibly triggers an entropy change of ~700 J kg$^{-1}$ K$^{-1}$, comparable to those of the commercial refrigerants in vapor-based compression systems. The Raman study and theoretical calculations reveal that applying pressure to the liquid state suppresses the twisting and random thermal motions of molecular chains, resulting in a lower configurational entropy. When the pressure is strong enough to drive the L-S-T, the configurational entropy will be fully suppressed and induce the colossal BC effect. This work could open a new avenue for exploring the colossal BC effect by evoking L-S-T materials.

[1] Key Laboratory of Materials Physics, Institute of Solid State Physics, HFIPS, Chinese Academy of Sciences, Hefei 230031, China. [2] Anhui Province Key Laboratory of Condensed Matter Physics at Extreme Conditions, High Magnetic Field Laboratory, HFIPS, Chinese Academy of Sciences, Hefei 230031, China. ✉email: tongpeng@issp.ac.cn; wjlu@issp.ac.cn; xlwang@theory.issp.ac.cn

Refrigeration technologies are widely applied in modern society, covering fields including food storage, air conditions, industrial manufacture, medical treatment, etc[1,2]. The vast majority of cooling equipment depends on conventional refrigeration technologies, which are based on compression cycles of powerful greenhouse gases, e.g., hydrofluorocarbons (HFCs) and hydrochlorofluorocarbons (HCFCs)[1–4]. These greenhouse gases have great global warming potentials thousands of times greater than $CO_2$ and a long atmospheric lifetime of several decades[1,3]. According to the Kigali Amendment, the phase-down of greenhouse gases is mandated[1,3,4]. Paradoxically, the demand for cooling will continue to increase in the seeable future because of population increase, urbanization, improvement in quality of life, etc. Although have been continuously optimized, environmentally harmful refrigerants seem unavoidable in present vapor-compression technologies[2,4]. Therefore, it is urgent to develop alternative cooling technologies, where the related refrigerants should be clean and environmentally friendly.

In recent decades, caloric cooling technologies based on the caloric effect have attracted great attention. The caloric effect refers to isothermal entropy change and adiabatic temperature change in response to a variety of external fields, and is often associated with the first-order phase transition in materials[5–7]. Depending on the types of external fields, such as the magnetic field, electrical field, uniaxial stress, and hydrostatic pressure, the caloric effects are sorted into magnetocaloric, electrocaloric, elastocaloric, and barocaloric (BC) effects, respectively[5–7]. Unlike the commercial refrigerants currently used in vapor-compression cycles, caloric materials have little environmental impact. To date, many achievements have been made in a large variety of solid-state caloric materials, where the isothermal entropy change driven by external fields is on the order of 10–200 J kg$^{-1}$ K$^{-1}$ [5–20]. Very recently, substantial progress was made in plastic crystals[21–24]. Reversible entropy changes ($\Delta S_r$) of ~470 and 515 J kg$^{-1}$ K$^{-1}$ were driven by ~580 and 570 MPa in NPA and NPG, respectively[23]. Although these $\Delta S_r$ values approach those of traditional refrigerants (e.g., $\Delta S_r$ ~ 520 J kg$^{-1}$ K$^{-1}$ for R134a)[21], it is a huge technical challenge to realize such high driving pressures in future cooling applications.

Matters can exist in different states, e.g., solid, liquid, and gaseous. When matter transforms from one state to another, the giant configurational change can be anticipated, leading to a colossal entropy change. Based on the gas–liquid transformation, traditional vapor-compression technology realizes the strong cooling capacity. Similar to the gas-liquid transitions, the liquid-solid-transition (L-S-T) also exhibits colossal entropy change[25–29]. This is why L-S-T materials are widely used in phase-change energy storage field[25–27]. Moreover, the L-S-T is often characterized by a large volume change since the density changes, which is thus sensitive to external pressure[29–31] and could be utilized for realizing colossal BC effects.

As typical L-S-T materials, $n$-alkanes (with the general formula $C_nH_{2n+2}$) are widely involved in human society, such as food preparation, biomedicine, cosmetics, mechanical and industrial fields, and many other uses. Here, we evaluated the colossal BC effect associated with the L-S-T around room temperature in $C_nH_{2n+2}$ with $n = 16$ and 18. Only a low pressure of ~50 MPa completely drives the $\Delta S_r$ of ~700 J kg$^{-1}$ K$^{-1}$, which is comparable to that of commercial refrigerants[21] and prominently larger than that of plastic crystals[21–24]. Moreover, a high adiabatic temperature change ($\Delta T_d$) of ~18 K was achieved under 100 MPa by direct measurement (i. e., direct method)[6], which doubled when the applied pressure was increased to 200 MPa. The excellent BC effect, particularly the colossal $\Delta S_r$ well exceeding those of the existing BC materials, suggests that $n$-alkanes are promising for caloric cooling applications. In the liquid state, $C_nH_{2n+2}$ exhibits a giant structural disorder due to the existence of a large number of gauche bonds. With increasing applied pressure, the population of gauche bonds decreases gradually until they basically disappear, which brings about a huge reduction in configurational entropy and leads to the colossal BC effect.

## Results

**BC performance for $C_nH_{2n+2}$ ($n = 16$ and 18).** The derived $\Delta S_r(T)$ are shown in Figs. 1a, b for $C_{16}H_{34}$ and $C_{18}H_{18}$, respectively (the details of measurement and data processing can be found in Supplementary Figs. 1–4 and Supplementary Note 1). Small pressures of approximately 58 MPa and 40 MPa activate $\Delta S_r$ values of 742 J kg$^{-1}$ K$^{-1}$ and 686 J K$^{-1}$ kg$^{-1}$ for $C_{16}H_{34}$ and $C_{18}H_{38}$, respectively. With increasing pressure, $\Delta S_r$ is increased slightly to 762 J kg$^{-1}$ K$^{-1}$ at 152 MPa for $C_{16}H_{34}$, and 711 J kg$^{-1}$ K$^{-1}$ at 232 MPa for $C_{18}H_{38}$. Here, only the L-S-T contribution was considered. And, $\Delta S_r$ will be enhanced if the additional entropy change is taken into account (see Supplementary Fig. 5 and Supplementary Note 1). The adiabatic temperature change $\Delta T_d$, i.e., the temperature jump in the $T(t)$ curves (see Supplementary Figs. 6, 7 and Supplementary Note 2), is shown in Figs. 1c, d for $C_{16}H_{34}$ and $C_{18}H_{38}$, respectively. For these two samples, when the pressure is too small to drive L-S-T, $|\Delta T_d|$ increases almost linearly with pressure (i. e., the incipient BC effect). When the pressure is larger than the critical value, the extra contribution from L-S-T comes in and results in an abrupt increase in $|\Delta T_d|$. In this case, a large $|\Delta T_d|$ can be achieved under low pressure as soon as the temperature is set slightly above the L-S-T temperature ($T_{LS}$). When the pressure is further increased to a higher value, a much larger $|\Delta T_d|$ is obtained. For example, $|\Delta T_d|$ reaches ~45 K under 300 MPa and ~57 K under 400 MPa. However, in the solid state, a pressure as high as 500 MPa can only induce a small $|\Delta T_d|$ (~10 K).

The maximum $|\Delta S_r|$ and $|\Delta T_d|$ values as a function of pressure are summarized and shown in Fig. 2a, b for $C_nH_{2n+2}$ ($n = 16$ and 18) along with those for the reported BC materials[9–13,16–19,23,32–39]. Here, the isothermal entropy changes in references are mainly obtained by the indirect method or quasi-direct method[6]. According to different treatment methods, either reversible entropy change ($\Delta S_r$) or nonreversible entropy change ($\Delta S_{ir}$) was obtained. In the literature, adiabatic temperature changes were measured either by the direct method ($\Delta T_d$), or by the quasi-direct method (reversible adiabatic

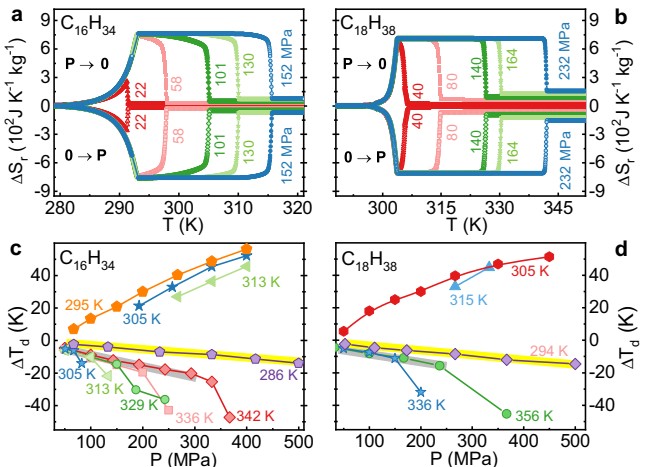

**Fig. 1 The reversible isothermal entropy change ($\Delta S_r$) and adiabatic temperature change measured by the direct method ($\Delta T_d$).** $\Delta S_r$ under different pressures for $C_{16}H_{34}$ (**a**) and $C_{18}H_{38}$ (**b**); $\Delta T_d$ values for $C_{16}H_{34}$ (**c**) and $C_{18}H_{38}$ (**d**).

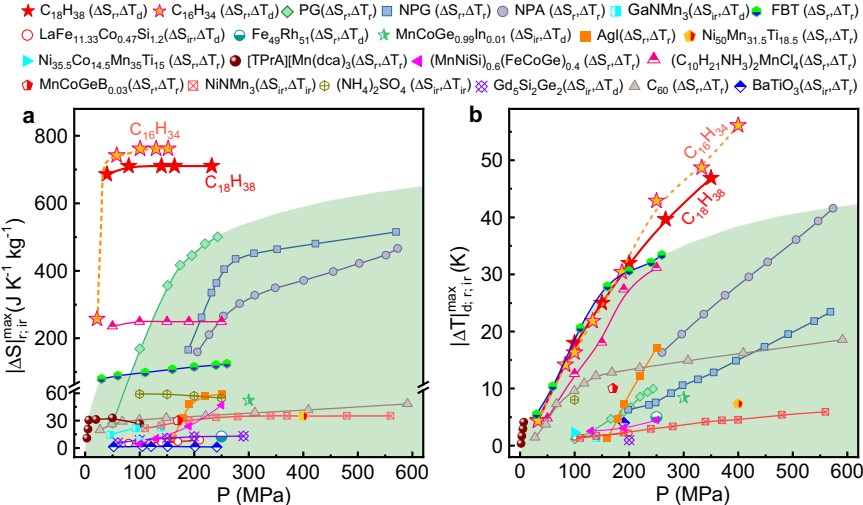

**Fig. 2 Comparison of BC performance between $C_nH_{2n+2}$ and the existing BC materials.** The maximum magnitudes of isothermal entropy change (**a**) and adiabatic temperature change (**b**) as a function of pressure for $C_{18}H_{38}$ and $C_{16}H_{34}$, are shown along with the reported BC materials. Here, the isothermal entropy change in references is mainly obtained by indirect or quasi-direct methods. According to different methods, reversible ($\Delta S_r$) and nonreversible entropy changes ($\Delta S_{ir}$) can be obtained in literatures. For the adiabatic temperature changes, some were measured by direct measurement ($\Delta T_d$). Others were estimated by the quasi-direct method, corresponding to either reversible ($\Delta T_r$) or irreversible values ($\Delta T_{ir}$). In general, $|\Delta S_{ir}|$ and $|\Delta T_{ir}|$, which ignore the inherent thermal hysteresis effect, are larger than $|S_r|$ and $|\Delta T_r|$, while $|\Delta T_d|$ is often less than the real value as it is difficult to obtain a fully adiabatic environment.

temperature changes $\Delta T_r$ and nonreversible adiabatic temperature changes $\Delta T_{ir}$[6]. For $C_{16}H_{34}$ and $C_{18}H_{38}$, $|\Delta T|_d^{max}$ (i.e., the maximum value of $|\Delta T_d|$) increases almost linearly with pressure up to 400 MPa and is larger than that of all reported materials excluding FBT[18], which exhibits comparable $|\Delta T|_r^{max}$ (i.e., the maximum value of $|\Delta T_r|$) values below 200 MPa. In regard to $|\Delta S|_r^{max}$ (i.e., the maximum value of $|\Delta S_r|$), $C_{16}H_{34}$ and $C_{18}H_{38}$ show better performance. Particularly in the low pressure range, only ~50 MPa can drive the $|\Delta S|_r^{max}$ of ~700 J kg$^{-1}$ K$^{-1}$, which already lies in the scope of the entropy change for commercial Freon-based refrigerants (e. g. R134a ~520 J kg$^{-1}$ K$^{-1}$)[21] and is at least three times larger than the reported values driven by the same pressure. As a comparison, even under pressures above 500 MPa, the $|\Delta S|_r^{max}$ values of plastic crystals (e.g., 470 J kg$^{-1}$ K$^{-1}$ at 580 MPa for NPA[23]) are still 30% less than the value of $C_{18}H_{38}$ at 40 MPa. Currently, when the pressure is approximately 50 MPa, the components used for standard hydraulics are available for future BC applications. Therefore, considering the refrigeration capacity and driving technology, the excellent low-pressure BC performance of $C_nH_{2n+2}$ indicates great potential as caloric refrigerants in the future.

When $n$-alkanes are in the solid state, the molecules exist mainly in the form of all-trans-type straight chains, which are arranged in an orderly manner because of the van der Waals force among the chains (Fig. 3a)[40]. After $n$-alkanes enter the liquid state, most C–C bonds (trans-type) rotate by 120° with the nearest C–C bond as the axis, forming gauche bonds. Consequently, the long molecular chains are twisted and turn into disordered gauche conformers[40]. For the alkane systems we studied, the large number of C atoms in the molecule causes hundreds of molecular conformers in the liquid state[41,42]. Therefore, it is difficult to accurately analyze the population of various conformers. As an alternative, the content of gauche bonds can be used as a parameter for evaluating the degree of molecular chain twisting, which is the key point for understanding the BC mechanism.

**Raman study under different pressure.** Raman spectroscopy is an effective probe to explore the lattice vibrations and internal interaction modes of molecules. Due to the pronounced

differences in energy between trans bonds and gauche bonds, it is possible to study their relative contents by Raman spectroscopy technology[40]. Figure 3b, c shows the variation of Raman spectra of $C_{18}H_{38}$ with temperature. When the temperature drops to 295 K, the characteristic bands of longitudinal acoustic modes at approximately 28, 62, and 134 cm$^{-1}$ appear, indicating the liquid to solid transition. Along with the structural evolution from disorder to order, the bands at ~1062 cm$^{-1}$ (the antisymmetric C–C stretching modes of all-trans bonds, $\nu_{as}(C–C)$)[43] and 1130 cm$^{-1}$ (symmetric C–C stretching modes of all-trans bonds, $\nu_s(C–C)$)[43] are significantly enhanced and sharpened. In contrast, the band at ~1080 cm$^{-1}$ (the C–C stretching modes of gauche segments, $\nu_g(C–C)$)[43] is almost completely suppressed. In addition, we found that the height ratio of the 1062 band to the 1080 band ($h_{1062}/h_{1080}$) increases gradually with decreasing temperature from 360 to 310 K (Fig. 3f), which means that cooling favors a less-disordered liquid state before the L-S-T happens. Similarly, the hydrostatic pressure also shows a regular influence on the liquid structure (as shown in Fig. 3d, e). When the pressure was increased from 0.1 to 210 MPa, $h_{1062}/h_{1080}$ increases from ~0.72 to ~0.81 (Fig. 3g). Obviously, applying pressure can effectively inhibit the formation of gauche bonds and straighten the molecular chains. A similar conclusion can be made by the molecular dynamics (MD) simulations as will be discussed in the following text. The MD results also give a good estimation of the evolutions of $T_{LS}$ and the entropy change at $T_{LS}$, which show a high consistency with the experiment results (Supplementary Figs. 8, 9a).

**Structural information calculated by MD simulations.** When the trans bond is converted into the gauche bond, the rotation process does not influence the C–C–C bond angle but brings about significant changes of the dihedral angles between the next-neighbor C–C bonds for $n$-alkanes[44]. As displayed in Fig. 4a, for solid $C_{18}H_{18}$ (e.g., at $T_{MD}^S = 300$ and 400 K, here $T_{MD}^S$ is the temperature used in MD simulation and higher than the real temperature due to the superheating problem[45]), the dihedral angles are mostly in the high-angle range of ~160°–180° due to the almost parallel alignment between the next-neighbor C–C

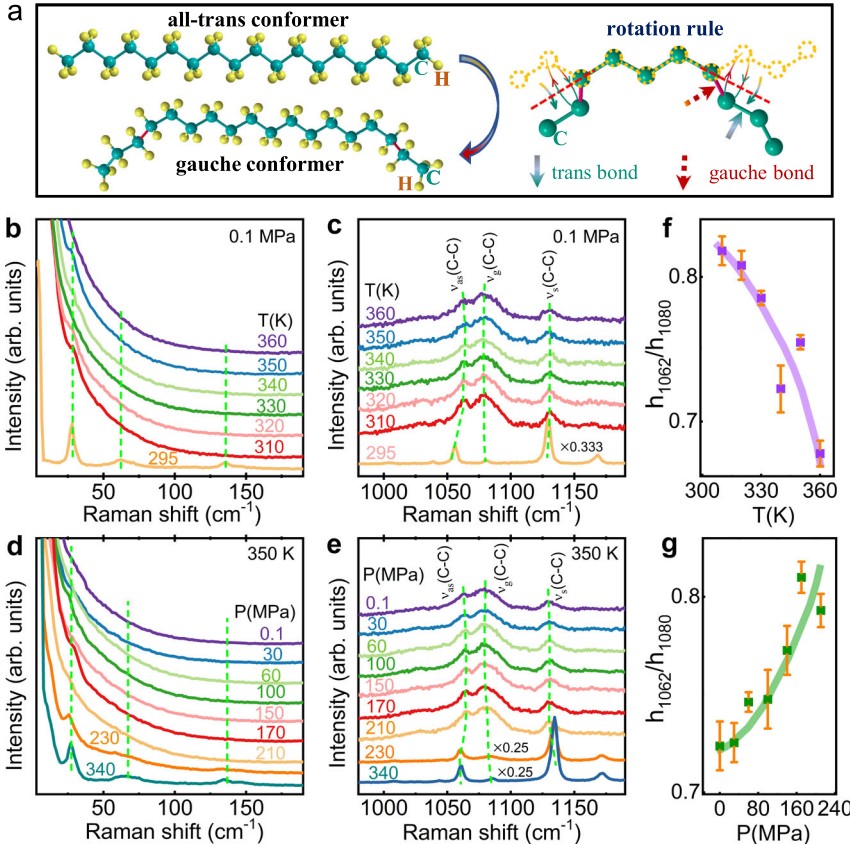

**Fig. 3 Raman spectra under different pressures at selected temperatures for C₁₈H₃₈. a** Schematic diagram of structural transformation between the all-trans conformer and gauche conformer. Temperature-dependent Raman spectra in the range Raman shift < 190 cm⁻¹ (**b**), and 980 cm⁻¹ < Raman shift < 1190 cm⁻¹ (**c**); pressure-dependent Raman spectra in the range Raman shift < 190 cm⁻¹ (**d**), and 980 cm⁻¹ < Raman shift < 1190 cm⁻¹ (**e**); the height ratio of the1162 band and 1180 band evolves with temperature (**f**) and pressure (**g**), where the orange error bars were displayed.

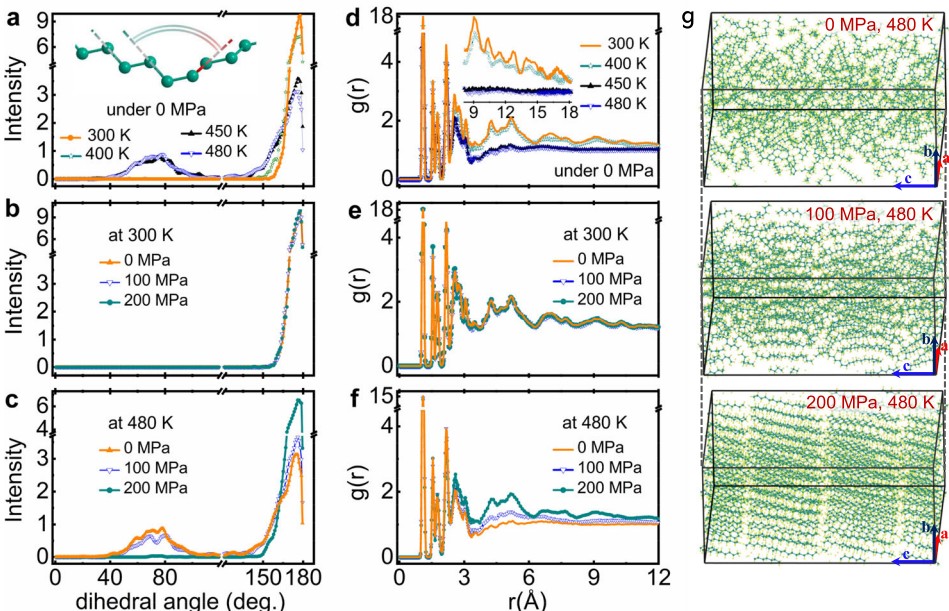

**Fig. 4 The theoretical calculation of structural information under different pressures for C₁₈H₃₈.** The distribution intensity of dihedral angle (sketched in inset of **a**) between the next-neighbor C–C bonds at different temperatures; the distribution intensity of dihedral angle under different pressures at $T^S_{MD}$ = 300 K (**b**), and $T^S_{MD}$ = 480 K (**c**); **d** the radial distribution function g(r) at different temperatures; the g(r) curves under different pressures at $T^S_{MD}$ = 300 K (**e**), and $T^S_{MD}$ = 480 K (**f**); **g** the calculated structures under 0, 100 and 200 MPa at $T^S_{MD}$ = 480 K.

bonds in the all-trans conformers. The weak deviation from the ideal 180° could be attributed to the thermal vibration. In the liquid state (e. g., $T_{MD}^S = 450$ and 480 K), a large number of gauche bonds appear, which destroys the parallel arrangement between the next-neighbor C–C bonds and results in the distribution of related dihedral angles in the low-angle range of ~50°−90°. Meanwhile, the distribution intensity of the dihedral angle in the high-angle range shows a significant decrease. Therefore, it is an effective way to grasp the evolution of the gauche bond population by analyzing the distribution of the C–C dihedral angles. As displayed in Fig. 4b and Supplementary Fig. 10a, the dihedral angle distribution in the solid state of $C_{18}H_{18}$ is nearly unaffected by the external pressure. In sharp contrast, when applying pressure to the liquid state of $C_{18}H_{18}$, the distribution intensity of the dihedral angle decreases in the low-angle range but increases in the high-angle range (Fig. 4c and Supplementary Fig. 10b). As the pressure reaches a critical value, the dihedral angles are only distributed in the high-angle range, indicating that $C_{18}H_{18}$ completes its transition from liquid to solid. This result again confirms that the pressure can promote the transformation from gauche bonds to trans bonds, which impedes the twisting of the molecular chains.

The radial distribution functions, g(r), of $C_{18}H_{18}$ under different temperatures and pressures are calculated and shown in Fig. 4d–f and Supplementary Fig. 11. For solid state $C_{18}H_{18}$, the g(r) curves show distinguishable peaks up to r = 18 Å, indicating the long-range structural order and low configurational entropy (Supplementary Fig. 9b). In the liquid state, the g(r) curves present a profile similar to that of the solid state in the range r < 3.6 Å, but do not show any peaks at longer distances. The g(r) curves of solid $C_{18}H_{18}$ are barely influenced under pressure and so are the calculated solid structures (Supplementary Fig. 12a, b). Upon applying pressure on the liquid $C_{18}H_{18}$, the g(r) peaks, especially in the range r > 3.6 Å, were enhanced gradually with pressure until the L-S-T is completed, indicating an enhanced structure ordering. Consistently, as intuitively displayed in Fig. 4g and Supplementary Fig. 12c, d, with the application of pressure in the liquid state, the molecular chains become straightened and their spatial arrangement is more ordered.

For present n-alkanes (i.e., $C_{18}H_{18}$ and $C_{16}H_{34}$), there exist hundreds of molecular conformers simultaneously in the liquid state due to the large number of C atoms. In addition to the thermal motions, numerous molecular conformers are constantly transforming among themselves, which causes a huge configurational entropy. Under pressure, the molecular spacing is decreased and the intermolecular force is strengthened, which not only reduces the thermal motion ability but also inhibits molecular twisting and thus reduces the number of active molecular conformers. Therefore, the configurational entropy decreases with increasing pressure, so that the incipient BC effect occurs. Accordingly, the molecular chains become straighter and arrange more orderly as pressure increases, as displayed by the static structure simulated by MD (Fig. 4g and Supplementary Fig. 12c, d). With further increasing pressure to the critical value, L-S-T occurs. All the molecular conformers will basically convert into the all-trans conformers, and the molecular thermal motions are completely suppressed. Consequently, the configurational entropy is fully suppressed, which drastically enhances the incipient thermal response, giving rise to the colossal BC effect.

## Discussion
The $T_{LS}$ of n-alkanes $C_nH_{2n+2}$ is strongly dependent on the n value and can range from ~247 to ~356 K when n increases from 11 to 40, which supplies a large temperature window for BC applications[26,28,29]. In addition, n-alkanes have high cycling

stability, advantages of self-nucleation to avoid supercooling, noncorrosiveness, long-term chemical stability without segregation, and commercial availability at reasonable costs[25]. Currently, n-alkanes, as the main component of paraffin, have been widely used for the storage of solar thermal energy in buildings, heat pumps, and spacecraft[25–27]. Many methods including the improvement of thermal conductivity and encapsulation have been proposed, which makes n-alkanes applicable to BC cooling similar to other solid compounds[25,26]. Meanwhile, we find that hydrostatic pressure is generated inside the n-alkanes when external pressure is applied to either liquid or solid states (see details from Supplementary Fig. 13). In this sense, an additional liquid pressure transfer medium can be avoided, and the thermal energy density can thus be increased in future BC cooling systems. Additionally, this feature makes it easy to improve the heat transfer rate of n-alkanes, for example, by embedding the copper foam into the pressure vessel (see Supplementary Fig. 14 for schematic diagram). By controlling the porosities of copper foam, the heat transfer efficiency of n-alkanes can be adjusted effectively.

In summary, we reported the colossal BC effect near room temperature in n-alkanes $C_nH_{2n+2}$ with n = 16 and 18. The n-alkanes exhibit extremely huge thermal responses to pressure near room temperature. Typically, at ~50 MPa the reversible entropy change reaches ~700 J kg$^{-1}$ K$^{-1}$, which is at least three times larger than those of other BC materials under the same pressure. As clarified by the Raman study and the theoretical calculations, in the liquid state the severe twisting and thermal motion of molecular chains are dramatically suppressed by applying pressure, decreasing the configurational entropy. As the applied pressure reaches the critical pressure that triggers the L-S-T, the incipient entropy change is strongly enlarged, resulting in the colossal BC effect. The excellent BC properties, tunable operating temperatures, lost-cost raw materials, and well-known thermal properties suggest n-alkanes are promising refrigerants for caloric cooling. This work also indicates that more excellent BC performance can be expected based on the L-S-T materials in the near future.

## Methods
**DTA measurement**. Calorimetric measurements under hydrostatic pressures were carried out using a home-built High-Pressure Differential Analysis (HPDTA) system. The similar equipment and related methods have been used in many previous reports[11,20,46,47]. One end of K-type thermocouple along with $C_nH_{2n+2}$ (MACKLIN, 99%) was sealed in a plastic capsule, and the other end along with the reference compound (i.e., $C_nH_{2n+2}$ with different n) was sealed in another capsule. Then they all were put into the DTA cell made of Teflon, using Daphne 7373 as the pressure medium. The Teflon cell was inserted in a Be–Cu-based pressure cylinder. The hydrostatic pressure was applied by the hydraulic machine. The annular heating sheet was adhered to the outer surface of the pressure cylinder, which is used for the temperature control in liquid nitrogen dewar. During the experiment, the heating and cooling rate was set at 1 K/min. The entropy change at the first order phase transition under ambient pressure was evaluated by the differential scanning calorimeter (DSC). Based on the DSC and DTA results, the entropy change of first-order phase transition under different pressures can be calculated[11,20,46,47].

**Adiabatic temperature change measurement**. When it comes to the direct measurement of the adiabatic temperature change, the same pressure apparatus used in DTA measurement was employed. Due to the high liquidity in liquid and extreme low modulus, the $C_nH_{2n+2}$ itself can serve as the pressure medium. So, one end of K-type thermocouple along with $C_nH_{2n+2}$ was sealed in the DTA cell made of Teflon, and the other end was embedded in the Be–Cu-based pressure cylinder. The measurements were carried out at the desired temperatures in cooling process and the data was collected in both pressure loading and unloading processes. A certain pressure was applied quickly to target values by manual means and maintained stably throughout the heat release cycle. Then the pressure relief value was opened and the pressure was released. The present hydraulic machine using the single-acting cylinder can realize the instant release of pressure, which is much faster than the pressure loading process.

**Raman spectra measurement**. The high-pressure condition was realized by employing a diamond anvil cell with anvils in 2000 μm culet and the high temperature condition was obtained using a specially designed cryostat with the temperature error less than 0.5 K. The solid $C_{18}H_{38}$ was sealed in the whole sample chamber with the diameter of 1000 μm. The pressure was determined using the ruby fluorescence method with considering the temperature correction. The Raman scattering spectra were obtained in a back-scattering configuration with a 488 nm sapphire laser beam. The excited power was less than 3 mW, and the laser beam was focused on the sample by a ×20 objective. The scattering Raman signal was collected by using a commercial Spectrometer designed by Princeton instruments.

**Theoretical calculation methods**. The classic molecular dynamics simulations were performed using the LAMMPS package with periodic boundary condition[48]. The adaptive intermolecular reactive empirical bond order (AIREBO) potential for a system of C–H atoms was used in our MD simulations[49]. The Newton equation of motion was integrated by the velocity Verlet algorithm and the time-step is 0.5 fs. The coexisting liquid–solid phases were used to simulate the melting temperature of $C_{18}H_{38}$, since the two-phase MD method is more accurate for the melting simulation[50,51]. To acquire the exact thermodynamic data of $C_{18}H_{38}$, one-phase MD method was employed and the related temperature $T_{MD}^S$ is larger than the real temperature due to the superheating problem[45]. Here the superheating of solids above their melting point exists in the perfect crystal without free surfaces or other defects, since in real materials melting is usually initiated at surfaces, grain boundaries, and other defects[45]. The constructions of the supercell and solid-liquid coexisting phase and the simulation processes are available in Supplementary Note 3. To estimate the lattice vibrational entropy of $C_{18}H_{38}$ in the solid phase, we used the quasiharmonic (QH) approach, a promising computational technique that has been used for investigations of caloric effects in mechanocaloric materials[52]. In the QH approach, the vibrational frequencies were first calculated and the vibrational entropy was subsequently obtained through the exact analytical expression (see details in Supplementary Note 3).

## Data availability

All relevant data that support the findings of this study are presented in the manuscript and supplementary information file. Source data are available from the corresponding author upon reasonable request.

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

## Acknowledgements
This work was supported by Key Research Program of Frontier Sciences, Chinese Academy of Sciences (CAS) (QYZDB-SSW-SLH015), National Key R&D Program of China under Contract No. 2016YFA0300404, National Natural Science Foundation of China (U1932127, 11774351, U2030114, 12004387), project No. TZ2016001 of Science Challenge and Users with Excellence Program of Hefei Science Center CAS (No. 2019HSC-UE008). The calculations were performed at Hefei Advanced Computing Center.

## Author contributions
P.T., J.C.L., and Y.P.S. conceived the study. J.C.L. planned the research and performed the characterization including DTA and adiabatic temperature change under different pressure. The HPDTA equipment used in this research was constructed under the joint effort of K.T., J.C.L., W.H.S., X.K.Z., and P.T. K.Z. and X.L.W. measured the Raman spectrum. W.J.L. performed the theoretical calculations. Results are discussed by J.C.L., P.T., W.J.L., and X.L.W. The manuscript was written by P.T. and J.C.L. All authors comment on the manuscript.

## Competing interests
The authors declare no competing interests.
