## [Peer Review File · Nature Communications]

Colossal and Reversible Barocaloric Effect in Liquid-Solid-Transition Materials n-alkanesREVIEWERS' COMMENTS

Reviewer #2 (Remarks to the Author):

In this work, Lin et al. analyse the barocaloric effects associated with the liquid-to-solid phase transition occurring in n-alkane compounds (C_nH_{2n+2}), which is very sensitive to pressure, by using different experimental techniques as well as atomistic molecular dynamics (MD) simulations. A reversible and colossal isothermal entropy change, ΔS , of the order of $700 \text{ J kg}^{-1} \text{ K}^{-1}$ is experimentally measured in $C_{16}H_{34}$ and $C_{18}H_{38}$, as a result of applying in them a pressure of around 500 bar near ambient temperature. The corresponding adiabatic temperature changes, ΔT , are also colossal, namely, of the order of 45-60 K. Raman spectroscopy and MD simulation analysis are carried out to characterise the pressure-induced liquid-to-solid phase transition in n-alkane compounds with atomistic detail.

The size of the barocaloric ΔS and ΔT proxies reported in this work for n-alkane compounds are truly colossal, and the overall level of characterisation and analysis for the pressure-induced phenomena underlying the reported barocaloric effects are sound and very satisfactory. Moreover, the possibility of tuning the critical temperature of the liquid-to-solid phase transition in n-alkane compounds in a totally rational and controlled way (i.e., by varying n) is actually a very interesting and potentially useful feature. For these reasons I do recommend publication of the present work in Nature Communications.

Nevertheless, I have few minor suggestions/comments which I would like the authors to address in order to further improve their article:

1. In the title, the authors write "low pressures". To me, in view of current state-of-the-art barocaloric materials and vapor-compression technologies, the ~ 100 bar that are necessary to fully drive the L-S transition leading to the reported giant barocaloric effects is not a small pressure (to be compared with ~ 10 bar used for state-of-the-art hybrid perovskites and involved in vapor-compression technologies). Therefore, I suggest to remove the adjective "low" from the title (although not from the rest of the text, in which it is made clear the order of magnitude of the pressures the authors are referring to).
2. A strong point of the present article is that atomistic simulations are presented in order to better understand the origins of the reported experimental giant barocaloric effects. The authors performed entropy change calculations based on the quasi-harmonic approximation for the analysed materials. This approach is very useful but probably not too well known in the context of caloric materials. In the recent review paper on mechanocaloric materials *Appl. Phys. Rev.* 6, 041316 (2019) the quasi-harmonic approach was mentioned and explained with certain detail, so that the authors should consider referring to that previous work in their manuscript. Also, the authors talk about superheating effects affecting their solid-liquid simulations. Since that concept may not be familiar to some potential readers, the authors should elaborate a bit more on that concept (e.g., by explaining in one or two sentences what they are about and probably also citing some previous related works in the literature like, for instance, *J. Chem. Phys.* 135, 024102 (2011)).
3. I find that the English grammar could be improved somewhat throughout the text. I acknowledge that this is a difficult task for non-native English speakers, but to the extent of the authors' possibilities that improvement should be attempted.

Response to Reviewer 2

Comment:

In this work, Lin et al. analyse the barocaloric effects associated with the liquid-to-solid phase transition occurring in n-alkane compounds (C_nH_{2n+2}), which is very sensitive to pressure, by using different experimental techniques as well as atomistic molecular dynamics (MD) simulations. A reversible and colossal isothermal entropy change, ΔS , of the order of $700 \text{ J kg}^{-1} \text{ K}^{-1}$ is experimentally measured in $C_{16}H_{34}$ and $C_{18}H_{38}$, as a result of applying in them a pressure of around 500 bar near ambient temperature. The corresponding adiabatic temperature changes, ΔT , are also colossal, namely, of the order of 45-60 K. Raman spectroscopy and MD simulation analysis are carried out to characterise the pressure-induced liquid-to-solid phase transition in n-alkane compounds with atomistic detail.

The size of the barocaloric ΔS and ΔT proxies reported in this work for n-alkane compounds are truly colossal, and the overall level of characterisation and analysis for the pressure-induced phenomena underlying the reported barocaloric effects are sound and very satisfactory. Moreover, the possibility of tuning the critical temperature of the liquid-to-solid phase transition in n-alkane compounds in a totally rational and controlled way (i.e., by varying n) is actually a very interesting and potentially useful feature. For these reasons, I do recommend publication of the present work in Nature Communications.

Nevertheless, I have few minor suggestions/comments which I would like the authors to address in order to further improve their article.

Our response:

Thanks to Reviewer 2 for his/her valuable comments on our manuscript, which is important for the manuscript to completely meet the standard of Nature Communications. Based on the Reviewer 2's suggestions, we have revised the manuscript. Following is our one-by-one response to the comments.

Comment 1:

In the title, the authors write "low pressures". To me, in view of current state-of-the-art barocaloric materials and vapor-compression technologies, the ~100 bar that are necessary to fully drive the L-S transition leading to the reported giant barocaloric effects is not a small pressure (to be compared with ~10 bar used for state-of-the-art hybrid perovskites and involved in vapor-compression technologies). Therefore, I suggest to remove the adjective "low" from the title (although not from the rest of the text, in which it is made clear the order of magnitude of the pressures the authors are referring to).

Our response:

Following Reviewer 2's comment, the content including "driven by low pressure" has been removed from the title in the revised manuscript.

Comment 2: A strong point of the present article is that atomistic simulations are presented in order to better understand the origins of the reported experimental giant barocaloric effects. The authors performed entropy change calculations based on the quasi-harmonic approximation for the analysed materials. This approach is very useful but probably not too well known in the context of caloric materials. In the recent review paper on mechanocaloric materials *Appl. Phys. Rev.* 6, 041316 (2019) the quasi-harmonic approach approach was mentioned and explained with certain detail, so that the authors should consider referring to that previous work in their manuscript. Also, the authors talk about superheating effects affecting their solid-liquid simulations. Since that concept may not be familiar to some potential readers, the authors should elaborate a bit more on that concept (e.g., by explaining in one or two sentences what they are about and probably also citing some previous related works in the literature like, for instance, *J. Chem. Phys.* 135, 024102 (2011)).

Our response:

In the revised paper, we have cited the references mentioned by Reviewer 2. Meanwhile, we have intensified the discussions related with "superheating effects" and "quasi-harmonic approximation" in "**Methods-Theoretical calculation methods**" section, shown as following: "The classic molecular dynamics simulations are performed using

the LAMMPS package with periodic boundary condition⁴⁸. The adaptive intermolecular reactive empirical bond order (AIREBO) potential for a system of C-H atoms is used in our MD simulations⁴⁹. The Newton equation of motion is integrated by the velocity Verlet algorithm and the time-step is 0.5 fs. The coexisting liquid-solid phases was used to simulate the melting temperature of C₁₈H₃₈, since the two-phase MD method is more accurate for the melting simulation^{50,51}. To acquire the exact thermodynamic data of C₁₈H₃₈, one-phase MD method was employed and the related temperature T_{MD}^S is larger than the real temperature due to the superheating problem⁴⁵. Here the superheating of solids above their melting point exists in the perfect crystal without free surfaces or other defects, since in real materials melting is usually initiated at surfaces, grain boundaries, and other defects⁴⁵. The constructions of the supercell and solid-liquid coexisting phase and the simulation processes are available in Supplementary Note 3. To estimate the lattice vibrational entropy of C₁₈H₃₈ in the solid phase, we use the quasiharmonic (QH) approach, a promising computational technique that has been used for investigations of caloric effects in mechanocaloric materials⁵². In the QH approach, the vibrational frequencies are first calculated and the vibrational entropy is subsequently obtained through the exact analytical expression (see details in Supplementary Note 3). "

Comment 3: I find that the English grammar could be improved somewhat throughout the text. I acknowledge that this is a difficult task for non-native English speakers, but to the extent of the authors' possibilities that improvement should be attempted.

Our response:

The English grammar of “the revised manuscript” has been checked and polished by using the AJE Digital Editing “AJE AI” (see Fig. R1), running by American Journal Experts which is a division of Research Square. Then, the revised manuscript was double-checked by Dr. Weiwei Zhao (annyzhao@ipp.ac.cn), who is the executive officer of International Science and Technology Communication of our institute (Hefei Institutes of Physical Sciences, CAS). Now we believe the revised manuscript is well written and can be easily understood by potential audiences.

Digital Editing

Thank you for using AJE Digital Editing! You now have access to this tool until 01 Jan 2022.

To begin, simply drag and drop your manuscript file to the area below, select the style of English, and click "Begin Editing." We will start editing your manuscript immediately, and your results will appear on this page within minutes.

Your Digital Edits

Document File (.doc or .docx)

 Drag and drop your file here or *select a file from your computer*

Style of English *

British English

American English

BEGIN EDITING

About AJE Digital Editing

The AI that powers AJE Digital has been trained on millions of manuscripts from more than 2000 areas of study, and each manuscript has been edited by some of the world's best professional editors. Unlike our competitors, who train their AI only on papers that have already been edited, we train our AI on both the original and edited files—allowing us to make up to 3 times more changes with over 95% accuracy.

Your Annual Plan

Your AJE Digital Editing plan is active until 01 Jan 2022. For questions about your plan or help with a file, please email support@aje.com.

?

Feedback

 Supplementary Information.docx

Edited on 2021-12-10 See details · Download edited file

 Manuscript.docx

Edited on 2021-12-10 See details · Download edited file

Fig. R1 The introduction of English polishing tool “AJE”. The “Manuscript” and “Supplementary Information” have been polished by “AJE”.